# Almost all working adults have at least one risk factor for non-communicable diseases: Survey of working adults in Eastern Ethiopia

**Aboma Motuma**[1]*, **Lemma Demissie Regassa**[2], **Tesfaye Gobena**[3], **Kedir Teji Roba**[1], **Yemane Berhane**[4], **Alemayehu Worku**[5]

**1** School of Nursing and Midwifery, College of Health and Medical Sciences, Haramaya University, Harar, Ethiopia, **2** Epidemiology and Biostatistics Department, School of Public Health, College of Health and Medical Sciences, Haramaya University, Harar, Ethiopia, **3** Department of Environmental Health Science, College of Health and Medical Sciences, Haramaya University, Harar, Ethiopia, **4** Department of Epidemiology and Biostatics, Addis Continental Institute of Public Health, Addis Ababa, Ethiopia, **5** Department of Epidemiology and Biostatistics, School of Public Health, Addis Ababa University, Addis Ababa, Ethiopia

* abomaabdi1@gmial.com

**Data Availability Statement:** All relevant data are within the paper and its Supporting Information files.

## Abstract

## Introduction

The disease burden and mortality related to Non-communicable Diseases (NCD) increased in the last couple of decades in Ethiopia. As a result, an estimated 300,000 deaths per annum were due to NCD. According to a World Health Organization report, 39% of the total deaths in Ethiopia were attributable to NCD. Rapid urbanization characterized by unhealthy lifestyles such as tobacco and/or alcohol use, physical inactivity, low fruits and vegetable consumption, and overweight drive the rising burden of NCD. However, studies on risk factors for NCD and associated variables are limited among working adults in Eastern Ethiopia. Therefore, this study aimed to examine the magnitude of the risk factors of NCD and associated factors among working adults in Eastern Ethiopia.

## Methods

A cross-sectional study was carried out among 1,200 working adults in Eastern Ethiopia that were selected using a simple random sampling technique from December 2018 to February 2019. Data were collected following the World Health Organization Stepwise Approach to NCD Risk Factor Surveillance (WHO STEP) instruments translated into the local language. A total of five risk factors were included in the study. The Negative Binomial Regression Model was used to determine the association between NCD risk factor scores and other independent variables. Adjusted incidence rate ratio (AIRR) with a 95% Confidence Interval (CI) was used to report the findings while the association was declared significant at a p-value of less than 0.05. STATA version 16.1 was used for data clearing, validating and statistical analysis.

**Funding:** The authors received no specific funding for this work.

**Competing interests:** The authors have declared that no competing interests exist.

## Results

Totally, 1,164 (97% response rate) participants were employed for analysis. Overall, 95.8% (95% CI: 94.4–96.7%) of the participants had at least one of the five risk factors of NCD. Furthermore, the proportion of participants that had all NCD risk factors was 0.3%. Among the participants, 47.5% were alcohol drinkers, 5.1% were current smokers, 35.5% were overweight, 49.1% exercise low physical activity, and 95% had less than five portions of fruits and vegetables intake per day. Higher risk factor scores were associated with those of advanced age (AIRR = 1.24; 95% CI: 1.01–1.53 in 35–44 age group and AIRR = 1.28; 95% CI: 1.01–1.62 in 45–54 age group), and the ones who are higher educational level (AIRR = 1.23; 95% CI: 1.07–1.43 for those who have completed secondary school and AIRR = 1.29; 95% CI: 1.11–1.50 for those who have completed college education).

## Conclusion

The overwhelming majority (95.8%) of the participants had at least one risk factor for non-communicable diseases. The risk score of non-communicable diseases was higher among those with advanced age and who completed secondary and above levels of education. In a nutshell, the finding shows the need for lifestyle modification and comprehensive non-communicable diseases prevention programs for working adults in Eastern Ethiopia.

## Introduction

The magnitude of NCDs increase throughout the world. Studies show that more than half of the burden of disease is attributed to the years lived with disability (YLDs) of NCD and injuries [1]. The burden of NCD is sweeping in the low-and middle-income countries (LMICs) and responsible for over 70% of deaths worldwide [2–4]. LMICs are dealing with the triple burden of infective and non-infective diseases in a poor environment, and resource constraints which is leading to a major concentration of risk for high mortality [4, 5]. In 2018, World Health Organization (WHO) reported that15 million people between the ages of 30 and 69 die from a NCD, and more than 85% of this "premature" death occur in LMICs [2].

Sub-Saharan Africa(SSA) hugely contributes to the global burden of diseases with disability-adjusted life years (DALYs) [6]. Studies indicated SSA had increased DALYs from 90.6 to 151.3 million between 1990 and 2017, i.e., 67% increase [7].This significant increase is partly due to lifestyle change and an increase in the aging population [8]. SSA countries contribute to 80% of the global years of lived disability due to the double burden of communicable and NCDs [6, 9]. There is no reason to doubt that changes in lifestyles and unplanned urbanization will have adverse health effects in SSA [6].

Similarly in Ethiopia, the disease burden and mortality related to NCDs is believed to have been increasing in the past couple of decades [10]. For example in Ethiopia, study showed that NCDs were the leading causes of age-standardized death rate, causing 711 deaths per 100,000 people and the DALYs has been increased from less than 20% to 69% between 1990 and 2015 [11]. Furthermore, an estimated of 300,000 deaths in 2016 [12] to 700,000 deaths occurred in 2018 due to NCDs [2]. Also, in Ethiopia, the burden of NCDs increased from 34% in 2014 [13] to 39% in 2018 [2]. After four years, in 2021, WHO estimated that the percentage of deaths from NCD was 42% of all the death [14].

Several studies revealed tobacco use, harmful consumption of alcohol, unhealthy diets, and physical inactivity are well recognized modifiable behavioral risk factors for NCDs [15–17]. The major risk factors are also likely to affect one or more of the other NCDs, and some of the NCD risk factors tend to appear in 'clusters' in individuals. Those appearing in clusters sets of risk factors account for a large fraction of the risk of NCD in the population [15, 18]. The reported leading risk factors included tobacco use, alcohol consumption, low fruit and vegetable intake, and physical inactivity [19]. This finding coincide with the fact that the lifestyle of working adults in Ethiopia has radically been changed in the last decade due to changing working environment, concentration of the middle-aged population, urban dwellers, risk of sedentarism, less physical activities on the workplace, and better access to technology and leisure lifestyle status [20, 21]. Consequently, the burden of risk factors of NCDs is on the rise despite contextual variation of the magnitude of the factors is inevitable. Therefore, empirically identifying the most common NCDs risk factors in a specific setting is important to plan appropriate prevention strategies [19, 20].

The government of Ethiopia has made a big step forward in the ratification and development of a national strategic action plan for the prevention and control of NCDs in 2014 [19]. This plan outlines the prevention of the major NCDs and their risk factors through comprehensive responses. However, empirical data on NCDs risk factors among working adults are limited in Ethiopia. Therefore, this study intended to examine the magnitude of NCDs risk factors such as smoking, alcohol consumption, physical inactivity, low fruits, and/or vegetable intake, overweight and their associated factors among employees of Haramaya University in Eastern Ethiopia.

## Materials and methods

### Study setting, design and period

This study was conducted with Haramaya University employees. The university is situated in eastern Ethiopia and has four campuses, eleven academic and research units, and four clinics which provide services to the university community. The university also runs a specialized referral hospital in Harar town that provides comprehensive health services to the general public. The university has about 7,176 employees of which 28.1% are female and 71.9% are male. In terms of job mix, 21.1% of the staff is academic and the others 78.9% are grouped under technical and administrative staff. The large size of the employees and the diversity of their jobs were among the reasons for choosing the university setting for this study. This study utilized a cross sectional study design and was conducted from December 2018 to February 2019.

### Population and selection criteria

The study population was the permanent employees of the Haramaya University, i.e., it excluded daily laborers, and all other non-permanent employees. The study involved employees who worked at least for 6 months in the University. Also, the critically ill, pregnant women, and those with some type of physical disability were excluded because of unsuitability for physical measurement.

### Sample size and sampling procedure

The study participants were selected from each unit proportional to the size of their respective department staff size. Within each unit, the participants were selected randomly using the payroll roster as the sampling frame. After assuming a 95% confidence level, and a 10% non-response rate, the sample size was calculated using OpenEpi 3.1. The same formula was

employed on a single population proportionally by considering the prevalence of each core risk factor in Nigeria [22]. For the second objective a double proportion formula we used to determine the sample size for significant factors as reported in previous studies [23]. After stratifying per the size of their respective department staffs, we included 1,200 employees from the nine colleges and one institute in the university using simple random sampling technique and each participant was selected based on the proportion to the size of their respective departments. Similar techniques of sampling and data collection procedures have been employed in the previous studies [24] (Fig 1).

## Data collection methods and data collectors

Data was collected by face-to-face interviews and physical measurement using structured questionnaires that adopted from the WHO-STEP survey instrument version 3.1. [25]. Experienced nurses and field supervisors who can speak the local languages fluently were recruited and trained for five days and used as data collectors. Anthropometric data (weight and height) were collected according to WHO-STEP wise approach for NCDs surveillance. Anthropometric measurements were carried out using standard procedures and calibrated instruments. Weight was measured with the participants' barefoot and wearing light clothes using a digital weight scale and measured to the nearest 0.1 kg decimal point. Height was measured using a stadiometer with the participant's shoes and any hats or hair ornaments removed. During height measurement, the

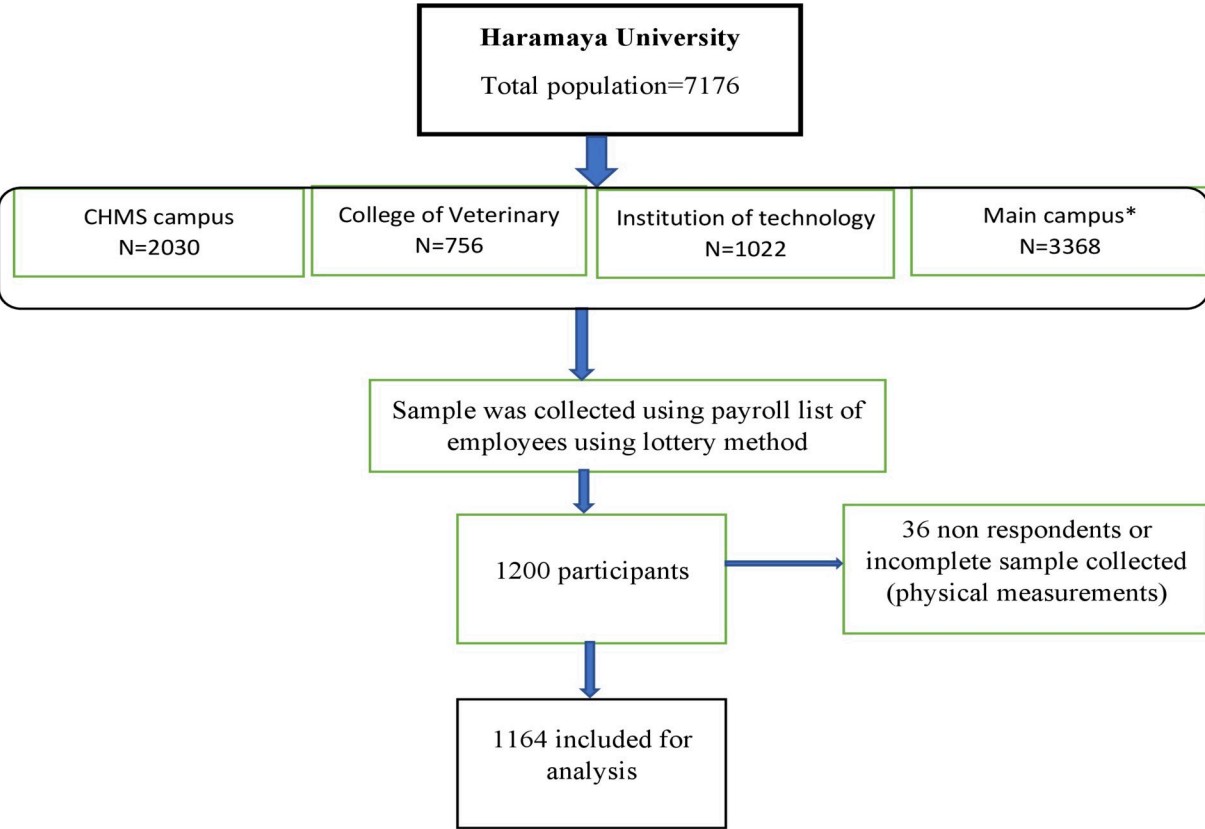

**Fig 1. Sampling procedures and response rate of study participants in Eastern Ethiopia, 2019 (n = 1164).** CHMS: College of Health and Medical Sciences. *: Seven colleges including: College of Agriculture and Environmental Science, College of Business and Economics, College of Computing and Informatics, College of Education and Behavioral Science, College of Law, College of Natural and Computational Science, and College of Social Sciences and Humanities, are found in the main campus.

participants faced away from the wall with their heels together and back as straight as possible. The head, shoulders, buttocks, and heels were in contact with the vertical surface. With the subject looking straight ahead, the head projection was placed at the crown of the head. The participant stepped away from the wall and the height measurement was recorded to the nearest 0.1 cm [25].

## Variables and measurements

In this study, the outcome variables are five core risk factors of NCDs (current tobacco and alcohol use, serving fruits and vegetables less than five times per day, low physical activity, and overweight) ranging from zero (having none of the risk factors) to five (having all the risk factors). The proposed explanatory variables in this study are socio-demographic factors such as age, sex, education, marital status, ethnicity, religion, occupation, year of service, and monthly income.

The questions to assess smoking status were categorized as never smoker, former smoker, and current smoker. *Current smoking* was defined as smoking every day or some days before a month of the data collection period. Then smoking status was coded as 'yes' if the participant has a history of smoking since the last one month before the survey period and ' no' if he/she has no history of smoking.

Similarly, the participant who took alcoholic drinks within 30 days preceding the study were asked if they consume any alcohols (beer, wine, spirits, and locally made alcoholic drinks like *beherawi teji, areke* or *katikala, tella,* and *bordee*). Respondents who reported 'Yes' were classified as current alcohol drinkers [26]. Nevertheless, participants who only consumed a few sips of alcohol during the past one month were categorized as 'no user.' We also asked them how frequently they had taken one standard alcohol drink (<1, 1–4, 5–6, 7 consumption days per week) [25]. To facilitate clear communication, the data collectors showed the respondents' pictures of alcoholic beverages as standard drinks and asked them (the respondents) the one they consumed.

Data was collected to seek the experience of the respondents as to the type of physical activity (work, leisure, and travel) they practice and the intensity levels (low, moderate and high). *High level*: physical activity items in the three domains i.e., of activity at work, travel, and recreational activities in a typical week were offered on the tool of data collection or questionnaire. A respondent is coded as *high level* when he/she exercises at least 3 days (vigorous-intensity activity) achieving a minimum of at least 1,500 metabolic equivalent minutes (MET-minutes) per week) or 7 or more days of any combination of walking (of moderate or vigorous-intensity activities) achieving a minimum of at least 3,000 MET-minutes per week. *Moderate level*: When a person exercises 3 or more days of vigorous-intensity activity for at least 20 minutes per day; or 5 or more days of moderate-intensity activity or walking for at least 30 minutes per day; or 5 or more days of any combination of walking, moderate or vigorous-intensity activities achieving a minimum of at least 600 MET-minutes per week. *Low level*: A person who does not meet any of the above-mentioned criteria falls in this category. Based on the above criteria, total physical activity per day was recorded taking into account all the three domains (work, transport and recreation-related activities).

Body mass index (BMI) was computed as weight (kg) divided by height (m) squared. Based on the WHO definition, BMI was grouped into four categories: underweight (BMI<18.5), normal weight (BMI = 18.5–24.9), overweight (BMI = 25.0–29.9), and obese (BMI≥30.0) [25].

Fruits and vegetables intake were asked for the number of days they ate fruits and vegetables in a typical week before data collection time. If they eat in a typical week, we were asked on average how many servings of fruits or vegetables they eat on one of those days. Servings were measured by demonstrating pictorial show cards. Eating less than five servings of fruits and/or vegetables per day is considered to be a low fruits and vegetable intake [25, 27].

Finally, the risk factors for NCDs were measured with a range of zero (no risk factors) to five (with all risk factors) scores.

## Data quality control

The original questionnaire was prepared in the English language and later translated into the local languages of Amharic and Afaan Oromo. Forward and backward translations were performed by two bilingual translators. Before data collection, we pre-tested the questionnaire in a similar setting and refined and validated it based on the feedback obtained during the pretest. Experienced nurses and field supervisors who can speak the local languages fluently were recruited and trained for five days and used as data collectors. The training was focused on the content of the questionnaire, data collection techniques, field procedures, and interviewing techniques. A field guide and data collection manual was used as a reference during the training. The field supervisors closely supervised the data collection processes and checked compliance with field procedures and the completeness of the questionnaires in the field.

## Data management and analysis

The completed questionnaires were double entered to EpiData Version 3.1 software and transferred to STATA 16.1 statistical software for analysis. The missing values of each variable were less than one percent. However, the chance of missing was unrelated to any variable or missing completely at random. Participant characteristics were described using proportion or mean based on their scale of measurement. We used Negative Binomial (NB) regression model analysis to examine the independent association between explanatory variables and the five scores of NCD risk factors, while controlling the confounding variables. We chose the Negative Binomial regression model because the assumption for ordinal logistic regression and the equal dispersion assumption for the Poisson regression model were not met. The Negative Binomial distribution has one parameter more than the Poisson regression that adjusts the variance independently from the mean of the outcome variable. Bivariate analysis was carried out between the risk factors of NCDs and the main independent variables. Multivariable Negative Binomial regression was executed to determine the association of independent variables with these five risk factor outcome variables. We used the k-fold cross-validation method to select an optimal model with the appropriate number of associated variables. The goodness of fit was determined using a likelihood ratio approach. Multicollinearity was checked using a correlation matrix and variance inflation factors for continuous variables. Also, the interaction between variables were checked and no significant interaction and severe collinearities were observed. Incidence rate ratio (IRR) with 95% confidence intervals (CI) were calculated. Statistical significance was declared at the 5% significance level (p-value <0.05).

## Ethical considerations

This study was conducted following the principles of the Declaration of Helsinki and the National Guideline for Ethics. The study protocol was approved by the Institutional Health Research Ethics Review Committee (IHRERC) of the Colleges of Health and Medical Sciences (CHMS), Haramaya University (Ref. No. IHRERC/196/2018).The written informed and voluntary consent was obtained from each study participant before data collection. Any data obtained from the participants was kept confidential and personal identifiers were removed from the sharable dataset. Furthermore, study participants with high-risk factors were linked to the university clinic for further consultation and screening.

## Results

### Socio-demographic characteristics of participants

From the total of 1,200 sampled participants, complete data of 1,164 participants were employed for analysis which made the response rate 97%. Female accounts 48.6% of the

respondents and the mean age of the participants (±SD) was 31.9 (±7.2) for the academic and 36.6 (±9.7) for support staff. Most of the participants, 722 (62%) were Orthodox Christians, 759 (63.5%) had a diploma or above educational level, and 667 (57.3%) were currently married. Eight hundred eighty-nine (76.4%) were supportive staff and 275 (23.6%) were academic staff. The mean (±SD) service year was 5.8 (±5.9) for academic and 8.0 (±7.9) for support staff. Of those who were able to estimate their earnings, the mean (±SD) reported per capita annual income of 105,059.1 (±49,960.38) Ethiopian Birr for academic staff and 39,473.3 (±35,246.49) Ethiopian Birr for support staff (Table 1).

## Magnitude of core risk factors

After clustering the patterns of the core risk factors, it was found that 4% (95% CI: 3.3, 5.6) of the participants were alcohol drinkers, physically inactive, overweight, and serving fruits or vegetable less than five times per day. On the other hand, 22.2% (95% CI:20.1, 24.9) had a cluster of risk factors of alcohol use, overweight, and serving fruits and vegetable less than 5 times per day. Moreover, more than 25.5% (95% CI: 23.4, 27.7) were physically inactive, overweight,

**Table 1. Socio-demographic characteristics of study participants in Eastern Ethiopia, 2019 (n = 1164).**

| Variables | Categories | Frequency | Percentage |
|---|---|---|---|
| Age category (year) | <25 | 80 | 6.9 |
| | 25–34 | 537 | 46.1 |
| | 35–44 | 324 | 27.8 |
| | 45–54 | 151 | 13.0 |
| | 55+ | 72 | 6.2 |
| Sex | Male | 598 | 51.4 |
| | Female | 566 | 48.6 |
| Ethnicity | Amhara | 549 | 47.2 |
| | Oromo | 509 | 43.7 |
| | Others [a] | 106 | 9.1 |
| Religion | Orthodox | 722 | 62.0 |
| | Muslim | 219 | 18.8 |
| | Protestant | 198 | 16.9 |
| | Others [b] | 25 | 2.3 |
| Marital status | Single | 427 | 36.7 |
| | Married | 667 | 57.3 |
| | Divorced | 49 | 4.2 |
| | Widowed | 21 | 1.8 |
| Educational status | Primary education | 193 | 16.6 |
| | Secondary education | 232 | 19.9 |
| | Diploma and bachelors | 559 | 48.0 |
| | Masters or beyond | 180 | 15.5 |
| Occupation | Academic | 275 | 23.6 |
| | Supportive | 889 | 76.4 |
| Average annual income (EB) | ≤15000 | 143 | 12.3 |
| | 15001–48000 | 513 | 44.1 |
| | >48000 | 508 | 43.6 |

EBR; Ethiopian Birr.

[a] ethnicity others: includes (Harari, Agaw, Wolaita, Hadiya, Shinasha, Kambata, Bench, Somali bench, Gamo and Gofa).

[b] religion others: includes Waqefata, Catholic, Adisawarya.

and consumed fruits and vegetable less than 5 times per day. In addition, 29.7% (95% CI: 26.3, 31.8) of the respondents had served fruits and vegetable less than 5 times per day and alcohol consuming, and overweight are the other co-risk factors. Likewise, 18% (95% CI: 17.3, 19.1) were overweight and fruits and vegetable served for less than 5 times per day, and 4.6% (95% CI: 3.1, 5.9) were smokers, and heavy or moderate drinkers.

The overall proportions of current smokers among the participants was 5.1% (95% CI: 3.9, 6.4), current alcohol drinkers was 47.5% (95% CI: 44.6, 50.4), being overweight was 35.5% (95% CI: 29.5, 38.4), low physical activity was 49.1% (95% CI: 46.2, 51.9), and serving fruits and vegetable less than five times per day was 95% (95% CI: 93.6, 98.5).

Besides, only 4.2% (95% CI: 3.3, 5.6) of the participants had none of the five risk factors and less than 0.3% (95% CI: 0.08, .0.8) had all risk factors for NCDs (Fig 2).

### Factors associated with the core risk factors

Using the score of NCDs risk factor as the dependent variable, we estimated the proportional incidence rate ratios for the demographic characteristics. Bivariate negative binomial models showed being older than 34 years, highly educated, married, serving more than ten years, and earning more than 6000 ETB monthly salary were significantly associated with having a higher rate score of NCD risk factors.

The multivariate analysis showed only age and educational level were associated with the highest level of risk factors of NCD. Age between 34–54 years was significantly associated with higher scores of NCD risk factors. Rates of having higher score of NCD risk factor increased by 24% (AIRR = 1.24; 95% CI: 1.01–1.53) and 28% (AIRR = 1.28; 95% CI: 1.01–1.62) among participants aged 35–44 years and 45–54 years, respectively, as compared to those below 25 years. Employees who had a secondary educational level had 23% (AIRR = 1.23; 95% CI: 1.07–1.43) higher rate of having a higher score of NCD risk factors. Meanwhile, the highly educated participants scored an increasing risk of having NCD risk factor. The employees whose educational level was diploma or bachelor degree had 29% (AIRR = 1.29; 95% CI: 1.11–1.50) higher rate of having a higher score of NCD risk factors than employees with primary educational level. Similarly, the odds of having a higher number of NCD risk factors increased by 26% (AIRR = 1.26; 95% CI: 1.02–1.56) among employees whose educational level was post-graduate (Master's or above) compared with those whose educational level was primary school (Table 2).

### Discussion

This paper examined factors associated with NCDs risk factors among working adult Haramaya University employees in Eastern Ethiopia. The magnitude of risk factors of NCDs in this study was high among working adults. The majority of employees have two or more of the five behavioral risk factors. Overall, low fruits and vegetable intake, alcohol use and physical inactivity were the most common risk factors. Age and educational levels were positively associated with the clustering risk factors of NCD. The implication of our finding indicates that the risk factors for NCD in this population are likely to increase with age. In the future, the risks of NCDs are probable to rise among the middle-aged working population, which has an impact on their productivity, unless they improve their lifestyle and appropriate intervention strategies are implemented. Combined, these unhealthy behaviors can have a significant influence on NCDs and life expectancy.

Among the risk factors for NCDs, low fruits and vegetables intake and alcohol use are the highest prevalent ones. This finding concedes with the previously conducted national survey in Ethiopia [26, 27], and other countries such as Vietnam [28]. Likewise, the prevalence rates

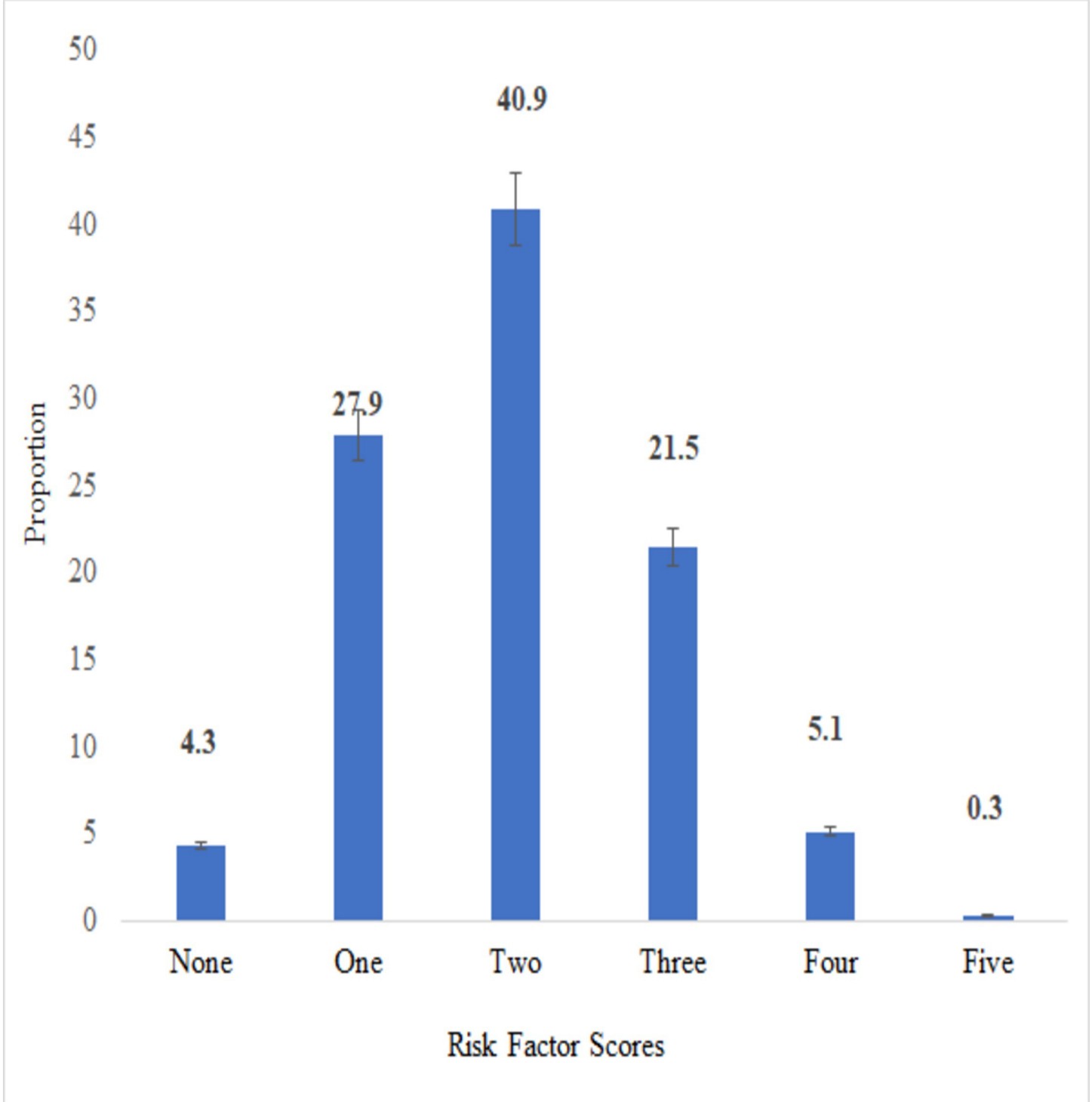

**Fig 2. The risk score of NCD risk factors among working adults in Eastern Ethiopia, 2019 (n = 1164).** Error bar shows a 95% confidence interval of the proportion of each risk score.

and cluster of behavioral risk factors among working adults were similar to those reported in other countries' surveys for different behaviors [22, 29–31]. As the data shows, among the risk

**Table 2. Multivariate for socio-demographic determinants for sum of risks factors of non-communicable diseases among working adults in Eastern Ethiopia, 2019.**

| Variable | Categories | CIRR | p-value | AIRR (95%CI) | P-value |
|---|---|---|---|---|---|
| Sex | Male | Ref | | Ref | |
| | Female | 0.98 (0.91, 1.07) | 0.68 | 1.02 (0.93, 1.12) | 0.666 |
| Age groups | < 25 years | Ref | | Ref | |
| | 25–34 years | 1.13 (0.94, 1.35) | 0.20 | 1.09 (0.91, 1.32) | 0.352 |
| | 35–44 years | **1.27 (1.06, 1.53)** | **0.01** | **1.24 (1.01, 1.53)** | **0.041** |
| | 45–54 years | **1.27 (1.04, 1.56)** | **0.02** | **1.28 (1.01, 1.62)** | **0.039** |
| | ≥55 years | 1.28 (1.01, 0.62) | 0.03 | 1.30 (0.99, 1.70) | 0.057 |
| Education | Primary | Ref | | Ref | |
| | secondary education | **1.20 (1.04, 1.38)** | **0.013** | **1.23 (1.07, 1.43)** | **0.005** |
| | Diploma and bachelors | **1.20 (1.06, 0.36)** | **0.004** | **1.29 (1.11, 1.50)** | **0.001** |
| | Post-graduate | **1.28 (1.10, 1.48)** | **0.001** | **1.26 (1.02, 1.56)** | **0.031** |
| Occupation | Academic | Ref | | Ref | |
| | Supportive Staffs | 0.96 (0.87, 1.06) | 0.42 | 1.03 (0.89, 1.19) | 0.712 |
| Marital status | Unmarried | Ref | | Ref | |
| | Married | **1.15 (1.06, 1.26)** | **0.001** | 1.08 (0.98, 1.20) | 0.122 |
| | Divorced | 0.99 (0.80, 1.24) | 0.97 | 0.98 (0.78, 1.24) | 0.872 |
| | Widowed | 0.90 (0.64, 1.27) | 0.41 | 0.90 (0.63, 1.28) | 0.557 |
| Service years | <5 years | Ref | | Ref | |
| | 5–10 years | 1.08 (0.98, 1.19) | 0.133 | 1.02 (0.92, 1.13) | 0.657 |
| | 10.1-15years | **1.17 (1.03, 1.33)** | **0.017** | 1.05 (0.91, 1.21) | 0.532 |
| | >15 years | **1.22 (1.07, 1.39)** | **0.003** | 1.07 (0.91, 1.25) | 0.413 |
| Monthly salary | <2000 ETB | Ref | | Ref | |
| | 2000–4000 ETB | 1.09 (0.98. 1.22) | 0.116 | 1.05 (0.93, 1.18) | 0.432 |
| | 4001–6000 ETB | 1.10 (0.96, 1.25) | 0.177 | 1.05 (0.90, 1.23) | 0.51 |
| | >6000 ETB | **1.23 (1.10, 1.37)** | **0.001** | 1.16 (0.99, 1.35) | 0.067 |

CIRR: Crude Incidence Rate Ratio; AIRR: Adjusted Incidence Rate Ratio; CI: Confidence Interval; ETB: Ethiopian Birr.

P-vales were calculated using the Negative Binomial regression model.

factors for NCDs, alcohol use, overweight, physical inactivity, and low fruits and vegetables serving were more prevalent risk factors, which is similar with the previously conducted national STEPs survey in Ethiopia [17, 26, 27] and other studies in different countries [28, 32–34]. Also, this study is consistent with the findings of other studies in Ethiopia [35]. The higher score of NCDs risk factors among working adults might be influenced by sedentary behaviors and low consumption of fruits and vegetables [20, 23]. In addition, working adults from higher socio-economic classes are known to adopt western lifestyles [36], which often causes a greater risk of the higher score of NCDs risk factors and greater intake of high fat and high caloric diet. This unhealthy lifestyle may substitute the healthy traditional diet (cereals, fruits, vegetables) [21, 37].

Moreover, the findings indicated that the proportion of overweight was much higher than that of the 2015 STEPS survey (men 4.4%; women 8.8%; urban areas 12.7%) [38] and that of WHO estimation (men 11.4%; women 28.3%) [2], and 30.1% for public employees in northern Ethiopia [20]. Although one possible explanation for the variation is the difference in the target population, the present finding quite convincingly indicates that overweight is rapidly increasing in the middle-aged working population in Ethiopia.

In congruence, nearly nine out of ten public employees in northern Ethiopia consumed less than the recommended amount of fruits and vegetables [20]. This might be because the

majority of the participants frequently consume animal products instead of vegetables and fruit. A survey in rural and urban Ethiopia also reported almost similar proportion (97.6%) as compared to the current finding (95%) [27]. Despite being the largest producer of fruits and vegetables, Ethiopia experienced a high population with a low intake of fruits and vegetables [20]. A higher level of inadequate intake of fruits and vegetables observed in the study is also consistent with findings from other LMICs [28, 39]. Except in two areas, namely Gilgel Gibe and Butajira, where about a quarter (27.0%) of the population is reported to be consuming fruits and vegetables, the rest of Ethiopia's population consumes these below adequate level (below five servings per day) [35]. One possible reason contributing for such a difference could be that Gilgel Gibe and Butajira are where the production of fruits and vegetables is better due to abundant rainfall. Fruits and vegetables are important components of a healthy diet. Reduced fruits and vegetables consumption is linked to poor health and increased risk of NCDs [27]. Addressing this situation requires multi-sectoral, and multi-pronged interventions to promote indigenous fruits and vegetables, and streamline its supply chain, which are crucial to enhance 'availability and affordability' of fruits and vegetables in Ethiopia.

The distribution of risk factors of NCDs was alarmingly high among the adult working force. This puts the working adults at higher risk of cardiovascular, strokes, and even cancer [20]. With a high level of NCDs risk factors, there is an increased risk of poor health outcomes. As several studies reported, the risk factors of NCDs are causally associated with deaths from NCDs, poor quality of life and, hence, poor productivity of working adults [28]. So, this finding suggests the need to promote and modify individuals' behavior, which in turn requires a multi-sectoral, and multidisciplinary public health responses. Moreover, an effective, realistic and affordable package of interventions and services should be strengthened to help people with risk factors of NCDs [39]. Regular health check-ups are known to be effective in detecting and preventing risk factors of NCDs at an early time before the occurrence of NCDs among working adults.

This study has found that older adults were at higher risk scores of the risk factors of NCDs compared with younger adults. The risk of having a higher score of NCD risk factors was significantly higher among people of 34–54 years of age. The association was also reported by previous studies conducted in Gilgel Gibe [35] and urban centers of southwestern Ethiopia [23]. This could be because they are less likely to adopt healthy behaviors. In other words, older adults experience little uptake of preventive measures as they tend to favor physical inactivity and sedentary life. Older adults might be physically inactive for a number of reasons. Some adults with NCDs become inactive because of the diseases [40, 41] and others may spend long hours sitting in offices, or watching television [42]. Moreover, the possible justification might be that as people get older and older the progressive reduction of the strength of musculature with age causes muscular atrophy, and decreased economic productivity and higher social isolation which in turn, lead to psychological problems. In other words, all these, in turn, might contribute to the risk factors of NCDs [37].

On the other hand, the higher educational level remained a major factor associated with higher scores of having NCD risk factors. This might be because of the fact that a working environment and working conditions influence the behavior of working adults. Interestingly, a higher educational level does not play a positive role in the reduction of risk factors. Educated people may have better socioeconomic status, urban life and higher economic status but these factors still contribute to risk factors of NCDs. This might, in turn, cause the likelihood of having a higher risk score. This coincides with a previous study from India which reported physical inactivity and overweight/obesity increased with increasing education levels [39]. Moreover, evidence from the 2015 National NCD survey in Ethiopia is in line with the result of this study [17]. Yet, the present finding contrasts with previous study which reported

prevalence of NCD was reversibly associated with educational level [43, 44]. The discrepancy might be due to the difference in the population of the study. Overall, the findings of this study indicate that prevention strategies for reducing risk factors of NCDs that include reducing alcohol consumption and smoking, promotion of physical activity and healthy diet may need to be developed separately for working adults.

Finally, this study has both strengths and limitations. The study is quite original in its attempts to explore the magnitude of NCDs risk factors among working adults, and predict the cluster core risk factors for the NCDs. Secondly, the study represents a comprehensive survey of NCDs risk factors using a standardized WHO-STEP wise approach for NCDs surveillance in developing countries. Data was collected by well-trained data collectors under close supervision. As such, it is a fairly comprehensive large-scale study using extensive sampling of an understudied setting, and adequate sample size.

This study relied on self-reported data through face-to face interviews on community sensitivity topics. Thus, social desirability biases (i.e., participants' tendency to report what is accepted in the communities) can't be ruled out and could result in underreporting of smoking and alcohol use. Recall bias as the questions on behavioral risk factors were self-report which is sometimes vulnerable to measurement errors which, in turn, might contradicts our expected estimation of magnitude and observed associations. Moreover, it should be noted that the study population was drawn only from one institution. In addition, this study did not address some risk factors of NCDs like environmental, biological and genetic risk factors. Therefore, the results of this study must be interpreted with caution.

## Conclusions

To conclude, the large majority (95.7%) of working adults had at least one risk factor for NCDs, and the most prevalent combination of risk factors were low fruit and vegetable intake, alcohol use, and physical inactivity. The risk score was significantly associated with older age, and higher level of education. Further research needs to be done to fully understand the trend and risk factors, preferably in a representative adult population. Lifestyle modification and comprehensive NCD prevention and control program is needed in higher learning institutions.

## Supporting information

**S1 Dataset.**
(DTA)

## Acknowledgments

We would like to thank Haramaya University and Addis Continental Institute of Public health for technical support; the study participants and data collectors for their kind cooperation.

## Author Contributions

**Conceptualization:** Aboma Motuma, Tesfaye Gobena, Kedir Teji Roba, Yemane Berhane, Alemayehu Worku.

**Data curation:** Aboma Motuma, Lemma Demissie Regassa, Tesfaye Gobena, Kedir Teji Roba, Yemane Berhane, Alemayehu Worku.

**Formal analysis:** Aboma Motuma, Lemma Demissie Regassa, Tesfaye Gobena, Kedir Teji Roba, Yemane Berhane, Alemayehu Worku.

**Funding acquisition:** Aboma Motuma, Kedir Teji Roba.

**Investigation:** Aboma Motuma, Kedir Teji Roba, Yemane Berhane, Alemayehu Worku.

**Methodology:** Aboma Motuma, Lemma Demissie Regassa, Tesfaye Gobena, Kedir Teji Roba, Yemane Berhane, Alemayehu Worku.

**Project administration:** Aboma Motuma.

**Resources:** Aboma Motuma, Kedir Teji Roba.

**Software:** Aboma Motuma, Lemma Demissie Regassa, Tesfaye Gobena, Kedir Teji Roba, Yemane Berhane, Alemayehu Worku.

**Supervision:** Aboma Motuma, Tesfaye Gobena, Kedir Teji Roba, Yemane Berhane, Alemayehu Worku.

**Validation:** Aboma Motuma, Tesfaye Gobena, Kedir Teji Roba, Yemane Berhane, Alemayehu Worku.

**Visualization:** Aboma Motuma.

**Writing – original draft:** Aboma Motuma, Lemma Demissie Regassa, Tesfaye Gobena, Kedir Teji Roba, Yemane Berhane, Alemayehu Worku.

**Writing – review & editing:** Aboma Motuma, Lemma Demissie Regassa, Tesfaye Gobena, Kedir Teji Roba, Yemane Berhane, Alemayehu Worku.

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
