## [Decision Letter · Decision Letter 0]

2 Dec 2021

PONE-D-21-30898Almost everyone has at least one risk factor for Non-Communicable Diseases - survey

of working adults in Eastern EthiopiaPLOS ONE

Dear Dr. Motuma,

Thank you for submitting your manuscript to PLOS ONE. After careful consideration, we feel that it has merit but does not fully meet PLOS ONE’s publication criteria as it currently stands. Therefore, we invite you to submit a revised version of the manuscript that addresses the points raised during the review process.

We look forward to receiving your revised manuscript.

Kind regards,

Wubet Alebachew Bayih, M.Sc.

Academic Editor

PLOS ONE

Journal Requirements:

● A clean copy of the edited manuscript (uploaded as the new *manuscript* file).

4. One of the noted authors is a group or consortium [Lemma Demissie Regassa, Tesfaye Gobena, Kedir Teji Roba, Yemane Berhane, Alemayehu Worku]. In addition to naming the author group, please list the individual authors and affiliations within this group in the acknowledgments section of your manuscript. Please also indicate clearly a lead author for this group along with a contact email address.

Additional Editor Comments:

General comments

Dear authors on your scholarly work; you have brought an important study problem with good findings that have public health importance in the area of practice. However, the manuscript has multiple language usage flaws including punctuations, wordings, spelling and mainly grammar errors. These problems are found throughout the manuscript. Therefore, please make repeated proof-reading and thorough copyediting before considering the manuscript for publication. This would help increase the readability of the manuscript if published.

Specific comments

Abstract

1.Background of the abstract doesn’t clearly show the existing burden of NCD in in Eastern Ethiopia or other regional states or the country. Generally, burden of NCD should first be stated followed by the objectives showing the research gap the authors would like to address.

2.Methods of abstract should include sampling technique, software for data entry and analysis, and cut off P-value to declare statistical significance of factors.

3.Results; kindly include response rate of the study at the beginning,.

4.Conclusion: The risk score of non-communicable diseases was higher for older and highly educated study participants…The phrase highly educated shall be clearly defined in the methods section. Moreover, kindly make your recommendation specific to your study area than considering the Ethiopian setting.

Methods

Population and selection criteria

5.Non-consenting individuals should have been considered as non-respondents than excluding them from the outset.

6.How did you identify severe mental disability?

7.Sample size and sampling procedure: It would be more self explanatory and easily understandable if the authors showed pictorial presentation (flow chart) of the sampling procedure including how many campuses �colleges � departments � sample size were considered to reach a response rate of 1,164 (97%).

8.Ethical clearance: What beneficent actions did the authors provide the employees (interviewees) in return for the interviews?

Discussion and conclusions

9.Recommendations should be specific and feasible in the given context of the study area.

Reviewers' comments:

Reviewer's Responses to Questions

**Comments to the Author**

1. Is the manuscript technically sound, and do the data support the conclusions?

Reviewer #1: Yes

Reviewer #2: Yes

2. Has the statistical analysis been performed appropriately and rigorously? 

Reviewer #1: Yes

Reviewer #2: Yes

3. Have the authors made all data underlying the findings in their manuscript fully available?

Reviewer #1: Yes

Reviewer #2: Yes

4. Is the manuscript presented in an intelligible fashion and written in standard English?

Reviewer #1: Yes

Reviewer #2: No

5. Review Comments to the Author

Reviewer #1: I am happy to review the manuscript entitled “Almost everyone has at least one risk factor for Non-Communicable Diseases - survey of working adults in Eastern Ethiopia” and I would like to thank the esteemed journal to invite me to review the manuscript.

The topic is very important indicating that the magnitude of risk factors of NCDs in the developing world. The aims of the manuscript are clearly defined with concise presentation of results. But I have some concerns that need clarification and which may improve the quality of the document.

1.To attract readers, I would like to suggest you to incorporate the “gap” under introduction of your abstract

2.Is that possible to indicate the available Ethiopian government policies/strategies towards NCDs like that of communicable diseases (under introduction part).

3.Under study setting of your methodology, you mentioned that “The university also runs a specialized referral hospital in Harar town that provides comprehensive health services to the general public” do found that the magnitude of risk factors of NCDs among health professionals is the same to the rest university employees.

4.Under your methods “Data were collected through face-to-face interviews and physical measurement using structured questionnaires adopted from the WHO-STEP survey instrument version 3.1” is the questionnaire validated in the local context or?

5.Under discussion: line 329 & 330“The magnitude of risk factors of NCDs in this study was high among working adults in Ethiopia”, how do you generalize a single institution finding for general Ethiopian working adults?

6.I suggest you to clearly indicate the general implication of your finding, under your discussion as an introduction of the discussion.

7.Figure 1: it is difficult to read the percentages from the figure

8.You have to introduce abbreviations when you use them for the first time (for instance “WHO STEP” under methods of the abstract.

Reviewer #2: General comment

1.The research has addressed very important and neglected public health problem in Ethiopia,; It benefits the current literature

2.The language needs revision

Comment

Title :: rewrite it as follows: Almost all working adults have at least one risk factor for Non-Communicable Diseases

Abstract

Method

WHO STEP in line 26 should be spell out before abbreviation.

Result

Line 33. Show the result with 95% CI

Rewrite this statement on line 37 and 38 as follow

“Higher risk factor scores were associated with advanced age (AIRR: 1.24; 95%CI: 1.01, 1.53 in 35-44 age group and AIRR: 1.28; 95%CI: 1.01, 1.62 in 45-54 age group), and high educational level (AIRR: 1.23; 95%CI: 1.07, 1.43 for those completed secondary school and AIRR: 1.29; 95%CI: 1.11, 1.50 for those completed college).

Conclusion section:

rewrite the sentence on line 43. “highly educated study participants” in more plausible words like those who completed high level education.

Introduction

The statement on line 58 &59 “This 67% increases in partly due to lifestyle change and ageing in sub-Saharan Africa” is not clear; re write it with more clear expression.

Methods

In line 152-15 what do you mean by co-variate? Are all mentioned co-variate were sued for analysis? If not why you name the as covariate. Age and service of year were indicated in this section; do not you think they can be correlated?

In line 153: ethnicity was mentioned as one of covariate. Using Ethnicity is not recommended to be used as covariate.

The statement under “Variables and measurements” is too long. Please only elaborate the measurement you used for your outcome. Avoid every detail in this section. If you are focusing on the current practice you should only elaborate in that context. Most of the statement and elaboration you made here are not useful.

Data management and analysis

This section lacks model fitness test with its statistic output, and you did not mention the multi-collinear variable. If you did not find any multi collinearity it is good to mention it.

Results

Line 252. Socio-demographic participant’s characteristics “re write as Socio-demographic characteristics participants”

Under socio demographic characteristics: avoid writing everything one by one; please focus on the important variable

Line 270: Legend EB, should be spell as ETB.

Line 275: Magnitude of core risk factors

It is highly recommended if you describe your result with 95% CI.

In this section what is the importance of table 2? Why you put your outcome variable with different variable? This section need more work. Please clearly show your outcome variable boldly. If you do not have a convincing reason delete table two and show your outcome variable in graph.

Line 305 Determinants of risk factors? Is determinant factor or associated factor is correct?

Please take care of the word you used when you explain the association. I urge you to see this section again.

6. PLOS authors have the option to publish the peer review history of their article (what does this mean?). If published, this will include your full peer review and any attached files.

Reviewer #1: No

Reviewer #2: **Yes: **Merga Dheresa

---

## [Author Response · Author response to Decision Letter 0]

24 Jan 2022

Thank you for indicating areas for improvement. Based on the editor and reviewers recommendations, we revised the manuscript and considered each and every comment and have responded point by point.

---

## [Decision Letter · Decision Letter 1]

16 Feb 2022

Almost all working adults have at least one risk factor for non-communicable diseases: survey of working adults in Eastern Ethiopia

PONE-D-21-30898R1

Dear Dr. Motuma,

We’re pleased to inform you that your manuscript has been judged scientifically suitable for publication and will be formally accepted for publication once it meets all outstanding technical requirements.

Kind regards,

Wubet Alebachew Bayih, M.Sc.

Academic Editor

PLOS ONE

Additional Editor Comments (optional):

Reviewers' comments:

Reviewer's Responses to Questions

**Comments to the Author**

1. If the authors have adequately addressed your comments raised in a previous round of review and you feel that this manuscript is now acceptable for publication, you may indicate that here to bypass the “Comments to the Author” section, enter your conflict of interest statement in the “Confidential to Editor” section, and submit your "Accept" recommendation.

Reviewer #1: All comments have been addressed

Reviewer #2: All comments have been addressed

2. Is the manuscript technically sound, and do the data support the conclusions?

Reviewer #1: Yes

Reviewer #2: Yes

3. Has the statistical analysis been performed appropriately and rigorously? 

Reviewer #1: Yes

Reviewer #2: Yes

4. Have the authors made all data underlying the findings in their manuscript fully available?

Reviewer #1: Yes

Reviewer #2: (No Response)

5. Is the manuscript presented in an intelligible fashion and written in standard English?

Reviewer #1: Yes

Reviewer #2: Yes

6. Review Comments to the Author

Reviewer #1: (No Response)

Reviewer #2: (No Response)

7. PLOS authors have the option to publish the peer review history of their article (what does this mean?). If published, this will include your full peer review and any attached files.

Reviewer #1: **Yes: **Tamirat Getachew

Reviewer #2: **Yes: **Merga Dheresa

---

## [Editor Report · Acceptance letter]

18 Feb 2022

PONE-D-21-30898R1 

Almost all working adults have at least one risk factor for non-communicable diseases: survey of working adults in Eastern Ethiopia 

Dear Dr. Motuma:

I'm pleased to inform you that your manuscript has been deemed suitable for publication in PLOS ONE. Congratulations! Your manuscript is now with our production department. 

Kind regards, 

on behalf of

Dr. Wubet Alebachew Bayih 

Academic Editor

PLOS ONE